# Atrial Fibrillation and Atrial Flutter Detection Using Deep Learning

**DOI:** 10.3390/s25134109

**Published:** 2025-07-01

**Authors:** Dimitri Kraft, Peter Rumm

**Affiliations:** 1MedTec & Science GmbH, 85521 Ottobrunn, Germany; 2custo med GmbH, 85521 Ottobrunn, Germany; peter.rumm@customed.de

**Keywords:** atrial fibrillation detection, 1D neural network, Holter monitoring

## Abstract

We introduce a lightweight 1D ConvNeXtV2–based neural network for the robust detection of atrial fibrillation (AFib) and atrial flutter (AFL) from single-lead ECG signals. Trained on multiple public datasets (Icentia11k, CPSC-2018/2021, LTAF, PTB-XL, PCC-2017) and evaluated on MIT-AFDB, MIT-ADB, and NST, our model attained a state-of-the-art F1-score of 0.986 on MIT-AFDB. With only 770 k parameters and 46 MFLOPs per 10 s window, the network remained computationally efficient. Guided Grad-CAM visualizations confirmed attention to clinically relevant P-wave morphology and R–R interval irregularities. This interpretable architecture is, therefore, well-suited for deployment in resource-constrained wearable or bedside monitors. Future work will extend this framework to multi-lead ECGs and a broader spectrum of arrhythmias.

## 1. Introduction

Atrial fibrillation (AFib) is one of the most common cardiac arrhythmias, posing significant challenges in the management of cardiovascular disease. AFib is the most frequently encountered arrhythmia, with an estimated prevalence of >3% in the adult population [1]. It is associated with increased morbidity and mortality with a 5-fold increased risk of stroke [2]. Its elusive and often asymptomatic nature makes timely detection and diagnosis crucial yet challenging. As the population ages globally, AFib is predicted to affect 6–12 million people in the USA by 2050 and 17.9 million in Europe by 2060 [3].

The advent of Holter monitoring in electrocardiography has opened new vistas in the continuous and noninvasive tracking of cardiac rhythms, presenting a potent tool in the early identification and management of AFib.

The significance of AFib lies in its association with an increased risk of stroke, heart failure, and other heart-related complications [4]. Traditional methods of AFib detection, primarily through standard resting ECG, often fail due to the intermittent nature of AFib episodes. In contrast, Holter monitors, capable of recording continuous ECG for extended periods, typically 24 to 48 h, provide a more comprehensive picture of a patient’s cardiac rhythm. This extended monitoring is pivotal in capturing transient episodes of AFib that could otherwise go undetected.

AFib can be categorized into three different types: paroxysmal, persistent, or permanent. Paroxysmal AFib is an episode that lasts seven days or less. Persistent AFib lasts more than seven days and requires additional therapy to end the episode, e.g., pharmacological or electrical cardioversion. In permanent AFib, therapy to cardiovert the rhythm is not attempted. Regardless of their duration, all three classes of AFib are associated with increased thromboembolic risks. Hence, accurate detection of AFib episodes, however transitory, and initiation of anticoagulation are key to minimizing downstream adverse events.

Systems developed for the automatic recognition of AFib primarily exploit two key features on ECG signals: absent P wave and/or irregular RR intervals. As such, the accurate detection of P or R wave peaks is critical. The ventricular rate is frequently fast, unless the patient is on AV nodal blocking drugs such as beta-blockers or non-dihydropyridine calcium channel blockers. Fibrillatory waves may or may not be detected. The low-amplitude P wave is especially susceptible to interference from ECG baseline drift and artifacts, which may lead to a degraded performance of P wave-based algorithms with noisy data signals [5].

This paper introduces a compact, end-to-end 1D ConvNeXtV2–inspired convolutional neural network (CNN) that detects AFib and AFL directly from raw single-lead ECG—no beat segmentation or handcrafted features required. We trained on a large, heterogeneous corpus (Icentia11k, CPSC 2018/2021, LTAF, PTB-XL, PCC 2017/2021, MIMIC-IV) and evaluated on MIT-AFDB, MIT-ADB, and NST. Our model achieved a state-of-the-art 0.986 F_1_ on MIT-AFDB while remaining lightweight (770 k parameters, 56 MFLOPs per 10 s window at 125 Hz). AFib and AFL were merged into a single class for simplicity. Guided Grad-CAM confirmed that the network attended to P-wave absence and R–R interval irregularities—key clinical markers. Against a 70 k-parameter 1D-CNN baseline, our approach not only outperformed but also matched much larger models [6], demonstrating robust generalization across datasets.

Our key contributions are as follows:A 770 k parameter 1D ConvNeXtV2 architecture for real-time ECG analysis.A multi-source training corpus ensuring broad generalization across recording conditions.Interpretability via Guided Grad-CAM, linking model focus to established AFib/AFL features.Benchmarking against a 70 k parameter 1D-CNN baseline and existing state-of-the-art methods.

## 2. Related Work

The detection of AFib using ECG data has been a focal point of research in the field of cardiac health monitoring, using advances in machine learning and deep learning techniques.

### 2.1. Consumer-Graded Devices on the Market

In one of the first studies using a smartwatch, researchers evaluated the Kardia Band (KB) attached to an Apple Watch for its ability to distinguish AFib from normal sinus rhythm (NSR) in 100 patients awaiting cardioversion (CV). They compared heart rhythms recorded by both a standard 12-lead ECG and the KB. Of 169 simultaneous ECG and KB recordings, 57 were unreadable by the KB. The KB showed a sensitivity of 93% and a specificity of 84% to detect AFib, with a K coefficient of 0.77. When physicians reviewed the KB recordings, sensitivity increased to 99%, specificity to 83%, and the K coefficient to 0.83. Blinded electrophysiologists achieved 100% sensitivity and 80% specificity with a K coefficient of 0.74 for unreadable KB recordings. There was also excellent agreement (K coefficient of 0.88) between KB and physician interpretations. The study concluded that the KB, particularly with physician review, can accurately differentiate AFib from NSR, suggesting its potential usefulness in screening patients prior to elective CV to avoid unnecessary procedures.

In a clinical validation study [7] of a smartwatch’s (Withings Scanwatch) single-lead ECG function to detect AFib, the authors found that while the device had a 14% rate of unreadable recordings, the cardiologist review reduced this to 4.1%. Automatic rhythm classification was less sensitive than manual interpretation, and the iECG’s automated function performed less effectively compared to other devices, although the cardiologist’s interpretation showed a high diagnostic accuracy of 98%. The study highlighted limitations, including testing only one device without comparisons, not assessing patient perception, and being a single-center study that may not reflect at-home usage.

The study by Fan et al. [8] evaluated the use of photoplethysmography (PPG) in mobile phones and smart bands for the detection of AFib, comparing it with standard 12-lead ECGs. Involving 108 inpatients, the research found high sensitivity (95.36%) and specificity (99.70%) in smart band PPG, with mobile phones also showing over 94% effectiveness in detecting AFib. Despite its promising results, the study had limitations like a small, specific patient group, a short testing duration, use of specific device models, and the exclusion of unclear cardiac rhythms, which might affect its generalizability.

The study by Mannhart et al. [9] aimed to evaluate the accuracy of five smart devices (Apple Watch 6, Samsung Galaxy Watch 3, Withings Scanwatch, Fitbit Sense, AliveCor KardiaMobile) in AFib against a physician-interpreted 12-lead ECG. Conducted at a tertiary referral center, it involved 201 patients, with AFib present in 31% of them. Sensitivity and specificity for AFib detection were similar between devices but varied slightly. Inconclusive tracings, where the device could not determine heart rhythm, were significant, ranging from 17% to 26% among the devices. A manual review of these tracings was often necessary, indicating the need for clinical supervision. The study highlighted that while these smart devices are useful, they have limitations in accurately detecting AFib without manual intervention, and that patient acceptance varied, with the Apple Watch being the most favored. Table 1 highlights the results of the study.

### 2.2. Innovative Monitoring Techniques

In a primary care setting, Chan et al. [10] evaluated the diagnostic performance of the Cardiio Rhythm smartphone app, a PPG tool, for the screening of AFib. The study, which involved patients with risk factors such as hypertension, diabetes, or aged > 65, compared the app’s performance against a single-lead ECG reviewed by cardiologists. The results showed that Cardiio Rhythm detected AFib with a sensitivity of 92.9% and a specificity of 97.7%, outperforming AliveCor’s automated algorithm in sensitivity but with a lower precision. Both methods demonstrated very high negative predictive values. The authors conclude that Cardiio Rhythm offers an accurate and reliable option for AFib screening in at-risk populations, suggesting its potential for widespread AFib screening.

Lahdenoja et al. [11] explored the innovative application of a smartphone’s built-in inertial measurement unit (IMU) for the detection of AFib through seismocardiography and ballistocardiography. In their approach, they extracted a range of spectral and time-domain features from the signals captured by the smartphone’s sensors. These features were then employed to train a classification algorithm. The study yielded promising results, demonstrating the potential of this method to facilitate convenient and accessible AFib detection for patients. This approach could significantly simplify the process for individuals to self-assess the presence of AFib, leveraging the ubiquity and technological capabilities of modern smartphones.

In their study, Izumi et al. [12] employed Microwave Doppler Radar as a novel approach for heart activity analysis. The researchers used an Autoencoder to reconstruct heart activity signals obtained from healthy individuals. The crux of their analysis involved a comparative examination of latent space representations derived from subjects diagnosed with AFib versus those with normal cardiac rhythm. By conducting an in-depth visualization of these latent space configurations, Izumi et al. were able to effectively segregate the data into two distinct clusters, corresponding to the aforementioned cardiac conditions. This demarcation serves as a significant step towards enhancing the diagnostic capabilities for AFib using advanced signal processing techniques in conjunction with machine learning methodologies.

Meghrazi et al. [13] made significant progress in the field of wearable health technology by creating a textile-based, multichannel ECG band designed for cardiac monitoring. This innovative band can effectively measure ECG signals from different positions around the waist. While the study did not specifically assess AFib detection, it successfully demonstrated the capability of textile sensors to acquire reliable ECG readings from the waist area. This development underscores the promise of integrating sensor fusion in wearable devices, paving the way for their potential commercial use in daily wear garments, facilitating continuous, noninvasive monitoring of heart rate variability.

Santala et al. [14] evaluated a consumer-grade, single-lead Necklace-ECG device for AFib detection. Tested on 145 patients, the device demonstrated high sensitivity (97.2–99.1%) and specificity (98.5–100%) in detecting AFib, as interpreted by cardiologists and an automatic AFib detection algorithm. These results suggest the Necklace-ECG as a feasible and reliable tool for AFib screening.

Conroy et al. [15] explored the detection of AFib using a noninvasive, inexpensive earlobe PPG sensor. Aimed at improving the detection of asymptomatic AFib, this technology is suitable for wearable devices, allowing for continuous cardiac monitoring. In a clinical trial, PPG recordings of AFib patients undergoing electrical cardioversion and a healthy control group were analyzed for heart rate variability indicative of AFib. The best-performing parameter yielded a sensitivity and specificity of 90.9%, showing that earlobe PPG signals can effectively match the results of costlier and more invasive ECG-based methods in AFib detection.

### 2.3. Overview of Performance

Most studies in the literature have focused on four different datasets to evaluate the performance of AFib detection algorithms:MIT-BIH Atrial Fibrillation Database (AFDB) [16],MIT-BIH Arrhythmia Database (ADB) [16],PhysioNet/CinC Challenge 2017 (PCC 2017) [17],MIT BIH Noise Stress Test (NST) [18].

Some datasets, such as PCC 2017, provide predefined test splits, whereas others—like the MIT BIH Arrhythmia Database—require users to define their own. Consequently, two main evaluation strategies have emerged: inter-subject splitting, in which training and test sets consist of disjoint subjects, and intra-subject splitting, in which different segments from the same individuals appear in both sets. Inter-subject evaluation offers a more stringent assessment of a model’s ability to generalize to truly unseen patients, while intra-subject evaluation typically yields higher performance figures at the cost of increased overfitting. This heterogeneity in splitting protocols makes direct comparisons across studies difficult, especially outside of challenge-specific benchmarks like PCC 2017, where test sets are fixed. Thus, reported sensitivities and specificities of 99% should be interpreted with caution, as they may not translate to genuinely novel data. An overview of AFib detection performance on PCC 2017 and the MIT BIH AFDB is presented in Table 2 and Table 3, respectively.

## 3. Methods

In the methodology section of this article, we dive into the specific techniques utilized for the classification of AFib periods in single-lead ECG data. Given the critical implications these heart rhythms have in clinical practice and patient care, the accuracy and efficiency of these detection algorithms are crucial.

### 3.1. Problem Formulation

We cast the task of classifying 1D ECG time series on a window-wise basis (30 s per window) as a binary classification problem. Let$$X=\{\;x_{1},x_{2},\ldots ,x_{n}\}$$
be the set of ECG windows, where each$$x_{i}=\left[\;x_{i}^{1},x_{i}^{2},\ldots ,x_{i}^{T}\;\right]$$
is a sequence of *T* samples (determined by the ECG sampling rate) spanning 30 s. We assign to each window $x_{i}$ a label$$y_{i}\in Y=\left\{0,\;1\right\},$$
where$$y_{i}=\left\{\begin{array}{cc} 1, & \mathrm{if}\;\mathrm{the}\;\mathrm{window}\;\mathrm{contains}\;\mathrm{AFL}\;\mathrm{or}\;\mathrm{AFib},\\ 0, & \mathrm{if}\;\mathrm{the}\;\mathrm{window}\;\mathrm{is}\;\mathrm{NSR}\;\mathrm{or}\;\mathrm{any}\;\mathrm{other}\;\mathrm{non}-\mathrm{AFL}/\mathrm{AFib}\;\mathrm{class}. \end{array}\right.$$

We choose a model $f_{\theta }:X\to \left\{0,1\right\}$ (e.g., a neural network) parametrized by θ. The parameters are learned by minimizing the binary cross-entropy loss$$L\left(\theta \right)\;=\;-\frac{1}{n}\sum_{i=1}^{n}\left[y_{i}\log f_{\theta }\left(x_{i}\right)\;+\;\left(1-y_{i}\right)\log\left(1-f_{\theta }\left(x_{i}\right)\right)\right]$$
over a training set. Our goal is to learn $f_{\theta }$ that accurately separates AFL/AFib windows (class 1) from NSR and all other rhythms (class 0).

### 3.2. Dataset Description

Previous studies have utilized a variety of ECG databases to train and evaluate CNN models, using intra- and inter-database ECG signals. A summary of the datasets is outlined in Table 4. We utilized the following datasets:**CPSC 2018:** Held in Nanjing, China, with two databases containing 13,256 ECGs. The recording varies from 6 to 144 s. Sampling frequency: 500 Hz. Training data [26].**CPSC 2021:** The data for this challenge were sourced from either 12-lead Holter or 3-lead wearable ECG devices, focusing on leads I and II from long-term ECGs, each sampled at 200 Hz. To ensure clear annotations, an AFib episode is defined with a minimum of 5 heartbeats. The first-stage training set includes 730 records from a mix of AFib and non-AFib patients, while the second stage has 706 records from a different patient group. The test set, drawn from similar and additional sources, includes data from diverse ECG systems and will not be released publicly. Training data [32].**St Petersburg INCART:** A total of 74 annotated ECGs from 32 Holter recordings, each lasting 30 min. Sampling frequency: 257 Hz. Training data [33].**Icentia11k:** The dataset segments each patient record into roughly 70 min segments, 50 segments randomly chosen per patient to maintain representation while reducing the size of the dataset. Data include patient-level information (3–14 days) that captures systematic features and segment-level data (around 1 h) that focus on disease indicators. The dataset contains 11,000 patients, over 2.7 billion labeled beats, and 541,794 segments. Specific focus is on AFib and AFlutter annotations, with AFib annotated 848,564 times and AFlutter 313,251 times in the dataset. The annotations are made at key points in the ECG rhythm, identifying normal and arrhythmic segments. Training data [34].**PTB:** Includes PTB and PTB-XL datasets with 22,353 ECGs. Recording lengths range from 10 to 120 s. Sampling frequencies: 500 or 1000 Hz. Training data.**Georgia Database:** Represents Southeastern USA, with 20,672 ECGs. Recording lengths: 5 to 10 s. Sampling frequency: 500 Hz. Training data [35].**Chapman–Shaoxing and Ningbo:** 45,152 ECGs, each 10 s long. Sampling frequency: 500 Hz. Training data [36].**UMich Database:** From the University of Michigan with 19,642 ECGs. Recording length: 10 s. Sampling frequencies: 250 or 500 Hz. Training data [36].**Long Term Atrial Fibrillation Database (LTAF):** Features 84 long-term ECG recordings of patients with paroxysmal or sustained AFib, typically spanning 24 to 25 h. Training data [37].**PhysioNet/CinC Challenge 2017 (PCC 2017):** Includes 8528 single-lead recordings from an AliveCor device, with durations between 30 and 60 s and various rhythm types. Validation data [17].**MIT BIH Atrial Fibrillation Database (AFDB):** The dataset comprises 23 records featuring dual ECG signals, with two exceptions (00735 and 03665) that only include rhythm and unaudited beat annotations. Each recording is 10 h long, digitized at 250 samples per second, with a 12-bit resolution within a ±10 mV range. These were recorded at Boston’s Beth Israel Hospital using ambulatory ECG recorders. The dataset includes rhythm annotations (.atr files) and beat annotations (.qrs files), the latter generated by an automated detector and not manually corrected. Some records have manually corrected beat annotations (.qrsc files). The beat annotations do not differentiate between beat types. Testing data [16].**MIT-BIH Arrhythmia Database (ADB):** The MIT-BIH Arrhythmia Database, assembled between 1975 and 1979, includes 48 half-hour, two-channel ECG recordings from 47 individuals. It is a mix of randomly selected and specifically chosen recordings from Beth Israel Hospital’s 24 h ambulatory ECGs, aimed at representing both common and rare arrhythmias. The data, digitized at 360 samples per second with an 11-bit resolution, were annotated by cardiologists, providing about 110,000 reference annotations. This complete database, partially available since 1999, offers 25 full records and annotations for all 48. Testing data [16].**CODE-15%:** This dataset provides a comprehensive collection of 12-lead ECGs with detailed annotations, encompassing a total of 345,779 exams from 233,770 patients. It represents a significant subset (15%) of the CODE dataset, specifically selected through stratified sampling. The data was collected by the Telehealth Network of Minas Gerais over a six-year period from 2010 to 2016. To ensure consistency, ECG signals of varying lengths (10 s at 4000 samples and 7 s at 2800 samples) were standardized to 4096 samples through zero-padding. This approach maintains the integrity of the data while allowing for uniform processing. Testing data [38].**SHDB-AF:** The dataset in this study was initially collected to evaluate the generalization performance of a deep learning algorithm, ArNet2 [39], for AFib detection. The data comprises ECG recordings from adult patients who underwent Holter monitoring between November 2019 and January 2022. These recordings were captured using a Fukuda Holter monitor at a sampling rate of 125 Hz, with two leads (modified CC5 and NASA) recorded over approximately 24 h per patient. While the dataset lacks reference beat annotations, each recording includes a diagnosis derived from the patient’s medical report. A total of 147 Holter recordings were collected, from which 100 recordings corresponding to 100 unique patients were selected for the study. Validation data [40].**MIMIC-IV:** The MIMIC-IV-ECG Diagnostic Electrocardiogram Matched Subset comprises approximately 800.000 ten-second, 500 Hz, twelve-lead ECG recordings from nearly 160.000 de-identified patients in the MIMIC-IV Clinical Database. Each recording is accompanied by machine-computed measurements and, where available, expert cardiologist reports, and can be linked back to the broader clinical dataset. In this work, we further filtered the collection to include only those waveforms whose diagnostic label contained either AFib or AFL, yielding our final cohort of atrial fibrillation/flutter examples [41].

### 3.3. Overview

The raw single-lead ECG signal is first divided into non-overlapping 10 s windows $x_{1},x_{2},\ldots ,x_{n}$. We adopt a single-lead approach because many consumer-grade devices (see Section 2.1) provide only one lead, and a model that requires fewer leads is more flexible; as such, it can be applied to single-, two-, or three-lead (Holter) recordings, whereas a twelve-lead requirement is far more restrictive.

Each 10 s window is then processed independently by our neural network classifier, which outputs a two-dimensional logit vector:$$\ell =\left[\ell_{\mathrm{NSR}},\;\ell_{\mathrm{AFib}}\right].$$

Applying the softmax function to *ℓ* yields the class-posterior probabilities$$\left[p_{\mathrm{NSR}},\;p_{\mathrm{AFib}}\right]$$
for each segment (see Figure 1).

### 3.4. Data Preparation

In our study, we used multiple preprocessing algorithms for ECG data to improve the detection of AFib/AFL. To maintain data integrity, records with less than 1250 data points (10 s at 125 Hz) are excluded. A critical step involves the standardization of diagnosis codes to ensure consistency across datasets (Sinus bradycardia, Sinus Tachycardia, Sinus Arrhythmia). Subsequently, each ECG channel undergoes a noise reduction cleaning process. The process involves a two-step filtering of the signal. Firstly, a high-pass Butterworth filter with an order of 5 and a cutoff frequency of 0.5 Hz is applied in a forward–backward manner. This is followed by a powerline filtering step. ECG signals were uniformly resampled to 125 Hz to match the input requirements of older Holter systems and to reduce computational and storage demands. While higher sampling rates (e.g., 250–500 Hz) can capture more detailed morphology and HRV features, the 125 Hz compromise provides an adequate resolution for P-wave and QRS analysis without unduly increasing model complexity or run-time [42].

### 3.5. HDF5 Dataset

We organized our data into an HDF5 dataset storing an N×C×L tensor, where *N* is the total number of ECG segments, C=1 corresponds to a single lead, and *L* is the segment length (10 s at 125 Hz). For training, we extracted 3,000,000 non-overlapping 10 s segments. To accommodate memory constraints, segments were lazy-loaded in batches during training.

Our training set was composed of segments from LTAF, Icentia11k, CODE-15%, INCART, PTB-XL, the Georgia 12-Lead Challenge, Chapman–Shaoxing and Ningbo, University of Michigan, PCC 2017, and MIMIC-IV-ECG. We reserved SHDB-AF for validation and evaluated the final performance on MIT BIH AFDB, MIT BIH ADB, and NST. This split strategy ensured a rigorous assessment of generalization to truly unseen data; in particular, the MIT BIH AFDB contains a wide variety of paroxysmal AFib signatures, providing a stringent test of model robustness.

### 3.6. Model Architecture

State-of-the-art models for single- or multi-lead AFib and AFL detection generally employ ResNet-style convolutional backbones, often augmented by an RNN-like layer just before the classification head [6,39,43]. To assess this approach, we used the ConvNeXt V2 architecture and benchmarked its performance against a simple, four-layer convolutional network with only 70 k parameters.

Our 1D-CNN is based on the ConvNeXt V2 framework [44], adapted for time series data classification (see Figure 2). The model comprises four stages of feature extraction, each containing multiple ConvNeXtV2-inspired blocks, with progressively increasing feature dimensions: 16, 32, 64, and 128.

The architecture begins with a stem layer that includes a 1D convolutional layer (kernel size of 4, stride of 4) followed by a LayerNorm [45]. Three additional downsampling layers, each combining a LayerNorm and a 1D convolution (kernel size of 2, stride of 2), progressively reduce the temporal resolution while increasing feature dimensionality.

Each feature extraction stage consists of several Block modules. A Block implements the following operations:**Depthwise Convolution**: A 1D depthwise convolution (kernel size of 7) captures temporal dependencies within each feature channel independently.**Normalization and Activation**: Features are normalized using LayerNorm and transformed via a GELU activation function.**Pointwise Convolutions**: A sequence of linear layers performs channel-wise transformations, first expanding the dimensionality by a factor of 4 and then reducing it back to the original size.**Global Response Normalization (GRN)**: This operation promotes inter-channel competition and enhances feature diversity.**Residual Connection**: The input is added back to the transformed output.

Following the feature extraction stages, global average pooling aggregates the temporal features into a fixed-length representation. A final LayerNorm layer and a fully connected linear layer produce the classification output. The model consists of 770 k parameters and does not consist of any RNN-like or transformer layers.

### 3.7. Augmentation

During the training phase, a strategic increase in data was carried out to promote a higher degree of generalizability and model resilience against variations in real-world scenarios. This augmentation process incorporated the random scaling of signal amplitudes, the infusion of random Gaussian, pink, and brown noise; the induction of minor baseline shifts, temporal shifting, negation of amplitude and adding negative artificial noise examples. In particular, we refrained from employing other commonly used augmentation procedures such as signal masking, time compression and stretching, mixup [46], and cutmix [47].

To evaluate the effect of data augmentation, we applied a combination of the following transformations dynamically during training:Scaling of the amplitude with a probability of p=0.75 and a scaling factor of [0.6,…,1.4];Offset of the amplitude with a probability of p=0.75 and an offset value of [−0.2,…,0.2];Addition of Gaussian, brown, or pink noise with a probability of p=0.75 and an offset value of [−0.2,…,0.2];Add time shift with a probability of p=0.75 and a shift value of [−625,…,625];Add baseline-wander with a probability of p=0.25;Flip the ECG by multiplying the amplitude by -1 with a probability of p=0.2;Use artificial noise as a hard example for non-AFib segments with a probability of p=0.05;

Although data augmentation does not invariably lead to enhanced performance in practical scenarios, certain methods can adversely affect results, as highlighted by Raghu et al. [48] in the context of AFib detection. However, based on our trials, our augmentation did improve the generalizability of our model. This aligns with Rahman et al.’s systematic review [49] on ECG data augmentation.

### 3.8. Post-Processing

After the model produces window-level probabilities via softmax, we translate them into a contiguous rhythm annotation and remove spurious predictions.


**Probability interpretation**


For each 10 s window, the softmax output$$\left[p_{\mathrm{NSR}},\;p_{\mathrm{AFib}}\right]$$
gives the estimated probabilities of NSR or AFib.


**Thresholding and smoothing**


Each 10 s window was labeled as AFib if the model’s predicted probability $p_{\mathrm{AFib}}\geq 0.75$; otherwise, it was labeled as NSR. To prevent rapid label alternations, adjacent windows with the same label were merged into continuous segments. The effects of this merging procedure and of varying the 0.75 threshold are evaluated in the Results Section.

## 4. Results

Results for our ConvNext V2 architecture model are outlined in Table 5 and Table 6. We calculate the metrics as follows for combined AFib and AFL duration as follows: The sensitivity and positive predictive value regarding combined AFib and AFL duration are calculated as follows:$$\mathrm{Sensitivity}=\frac{\mathrm{Overlap}\;\mathrm{of}\;\mathrm{predicted}\;\mathrm{AFib}/\mathrm{AFL}\;\mathrm{duration}}{\mathrm{Reference}\;\mathrm{AFib}/\mathrm{AFL}\;\mathrm{duration}}$$$$\mathrm{Precision}=\frac{\mathrm{Overlap}\;\mathrm{of}\;\mathrm{predicted}\;\mathrm{AFib}/\mathrm{AFL}\;\mathrm{duration}}{\mathrm{Predicted}\;\mathrm{AFib}/\mathrm{AFL}\;\mathrm{duration}}$$

The ConvNext V2 model achieved an overall sensitivity of 0.968, precision of 0.944, and F_1_ score of 0.957 on the MIT ADB dataset. Record 222 remains the outlier with lower sensitivity (0.593).

The ConvtNext V2 model achieved an overall sensitivity of 0.982, precision of 0.990, and F_1_ score of 0.986 on the MIT AFib DB. Records 4015 (precision 0.298) and 5091 (sensitivity 0.543) remain outliers.

### 4.1. Comparison with a 1D CNN Baseline

To evaluate the performance of the ConvNextV2 model, we compare it against a simpler baseline: a four-block one-dimensional convolutional neural network (1D CNN). Each block of this baseline model consists of a strided convolutional layer, followed by batch normalization, dropout (*p* = 0.25), and a ReLU activation function. The convolutional layers use a kernel size of 7, a stride of 2, and the same padding. The classification head integrates global average pooling and global max pooling, followed by a fully connected linear layer for final classification.

We assessed both models on the MIT AFDB and MIT ADB datasets, measuring sensitivity, precision, and F1-Score. The results (Table 7) are summarized below:

On the MIT AFIB dataset, ConvNextV2 achieved a sensitivity of 0.982, precision of 0.990, and F_1_ score of 0.986, outperforming the 1D CNN baseline (0.948, 0.992, 0.970). On the MIT ADB dataset, ConvNextV2 attained (0.982, 0.934, 0.958) versus the baseline’s (0.964, 0.960, 0.962), giving the baseline a slight F_1_ edge despite ConvNextV2’s higher sensitivity.

### 4.2. Evaluation Method

To evaluate our performance, we first merge AFib Regions to avoid unrealistic sequences. After merging, we calculate the total overlap between the ground truth and prediction. This total time overlap is used to determine sensitivity and precision. We further evaluate the impact of the merging and classification threshold.

#### 4.2.1. Calculate Overlap

Given a set of reference AFib and AFL regions $R=\{r_{1},r_{2},\ldots ,r_{m}\}$ and a set of predicted AFib and AFL regions $P=\{p_{1},p_{2},\ldots ,p_{n}\}$, where each region $r_{i}$ in *R* and $p_{j}$ in *P* is defined by its start and end times, denoted as $r_{i,\mathrm{start}},r_{i,\mathrm{end}}$ and $p_{j,\mathrm{start}},p_{j,\mathrm{end}}$, respectively, the overlap between a single reference region $r_{i}$ and a predicted region $p_{j}$ can be calculated as follows:$$\mathrm{Overlap}\left(r_{i},p_{j}\right)=\max\left(0,\min\left(r_{i,\mathrm{end}},p_{j,\mathrm{end}}\right)-\max\left(r_{i,\mathrm{start}},p_{j,\mathrm{start}}\right)\right)$$

The total overlap between all reference and predicted regions is then the sum of all individual overlaps:$$\mathrm{TotalOverlap}=\sum_{i=1}^{m}\sum_{j=1}^{n}\mathrm{Overlap}\left(r_{i},p_{j}\right)$$

This formula ensures that only positive overlaps (where the start of the overlap is less than the end) contribute to the total overlap, with zero used to handle cases where there is no overlap, preventing the calculation from producing negative values.

#### 4.2.2. Merging of AFib Regions

Since we trained our model with 10 s, single-lead ECG windows, we apply a post-processing merging step (gap closing). Per the 2020 ESC guidelines, an AFib episode must persist for at least 30 s to be formally diagnosed [50]. However, window-based AFib detection can artificially fragment continuous AFib episodes due to occasional P-wave artifacts or brief rhythm variations within what cardiologists consider a single episode.

Our merging approach addresses this technical limitation by combining adjacent AFib intervals separated by short gaps of non-AFib. After thresholding each 10 s window’s softmax output (e.g., $p_{\mathrm{AFib}}>0.7$) to obtain a binary AFib/NSR label sequence, we merge AFib regions based on a gap threshold $T_{\mathrm{gap}}$. Concretely, letL=[AFib,AFib,NSR,NSR,NSR,AFib]
denote six consecutive 10 s windows. With a merge-gap threshold of 30 s (i.e., up to three consecutive non-AFib windows), we treat the final AFib window as contiguous with the first two, yielding one continuous AFib segment of length 6×10 s = 60 s. In general, any two AFib runs separated by at most $T_{\mathrm{gap}}$ seconds of NSR are merged into a single AFib episode spanning from the start of the first to the end of the last window.

For example, consider a scenario where

**Window 1 (0–10s):** AFib detected;**Window 2 (10–20s):** NSR detected (due to occasional P-wave artifacts);**Window 3 (20–30s):** AFib detected.

Without merging, this yields two 10 s episodes (both <30 s, thus, not clinically significant). With merging, this correctly identifies a single 30 s episode meeting clinical thresholds. Other approaches to smooth the binary sequence predictions might contain a moving average [51] or sliding window [52].

We optimized $T_{\mathrm{gap}}$ empirically, finding that a 40 s gap threshold achieved an optimal F_1_ score on MIT-BIH AFDB (Table 8; Figure 3 and Figure 4). While merging increases sensitivity substantially, precision decreases minimally, indicating improved performance for estimating total AFib burden.

## 5. Model Interpretability

To uncover which portions of the ECG signal drive our network’s decisions, we apply Guided Grad–CAM [53]. Guided Grad–CAM generates a heatmap over the input time series, where warmer colors indicate stronger contributions to the AFib class (see Figure 5 and Figure 6). This visualization allows for linking the model activations back to clinically meaningful waveform features.

In the ECG domain, interpretability is essential: clinicians rely on specific signal characteristics (e.g., P-waves, R–R intervals) when diagnosing AFib [50]. By inspecting our Guided Grad–CAM outputs, we confirmed that the model indeed concentrates on the same signal components experts use. In particular, we observed the following:**P-wave abnormalities.** Heatmaps consistently highlight regions where the P-wave is absent, flattened, or morphologically distorted.**Irregular R–R intervals.** The model attends to segments with variable beat-to-beat timing, corresponding to an irregular rhythm of AFib.

## 6. Discussion

We evaluated our ConvNeXt-v2 model on the MIT-AFDB dataset by varying three hyperparameters: the ECG window length, the segment-merging threshold, and the post-softmax AFib probability cutoff. Our experiments show that a 10 s window, a 40 s merging threshold, and a probability cutoff of 0.7–0.8 maximize the F1-score, with the merging threshold exerting the greatest influence on the precision–sensitivity trade-off. Larger input window sizes led to poorer performance, likely because the model was trained exclusively on 10-second ECG segments (see Table 8).

Under these settings, ConvNeXt-v2 (770 k parameters) achieved an F1-score of 0.986 —matching the state of the art in [6], despite that model containing around 6M parameters (estimated). Relative to a baseline 1D-ConvNet (70 k parameters), ConvNeXt-v2 yielded a 1.86 pp F1-Score improvement on AFDB by scaling the model size by an order of magnitude. On the MIT-ADB cohort, however, no performance gain was observed, likely due to the limited number of AFib examples.

Furthermore, ConvNeXt-v2 requires only ≈46.3 MFLOP per 10 s window at 125 Hz (or ≈16.7 GFLOP per hour of data, as estimated using fvcore [accessed 26 May 2025]). On a modern CPU capable of ∼60 GFLOP/s, one second of compute can process ≈3.6 h of single-lead ECG, making ConvNeXt-v2 well suited for real-time inference on resource-constrained devices and Holter monitoring.

### 6.1. Performance on MIT ADB

Table 5 summarizes the per-record performance of our ConvNeXt-V2 model on the MIT ADB dataset (records without any true or predicted AFib intervals are omitted). Overall, the model achieved a sensitivity of 0.968 and a precision of 0.944. For records that contain no AFib episodes (e.g., 100–124, 200–223), the model correctly produced zero detections, demonstrating its ability to avoid false positives in NSR. While most recordings exhibit strong agreement between predicted and true AFib durations, the variability across a few records suggests that further refinement of temporal localization (onset) is warranted to ensure uniformly robust performance.

### 6.2. Performance on MIT Noise Stress Test DB

The MIT Noise Stress Test DB does not contain any AFib episodes, and the model’s zero sensitivity and precision values across all records are consistent with this fact. This outcome confirms that the model does not produce false positives in noisy datasets where no AFib is present. While this demonstrates good specificity, further validation is required to assess the model’s robustness in scenarios where noise is present alongside AFib episodes.

### 6.3. Performance on MIT AFDB

Table 6 reports our results on the MIT AFDB dataset, demonstrating consistently high performance across most records. A few outliers, specifically records 4015 and 5091, exhibit reduced precision and sensitivity, respectively. However, these recordings contain only a few minutes of AFib, so their contribution to the overall metrics is limited. In most cases, only a handful of AFib episodes are not detected correctly, indicating that future work should focus on improving the temporal localization of arrhythmia events.

### 6.4. Future Directions

Future work should focus on enhancing the model’s generalizability and robustness. For datasets where no AFib episodes are present, the model already demonstrates good specificity. However, efforts should be made to improve the consistency of its performance across records where AFib is present. Incorporating advanced temporal modeling techniques and refining the feature extraction process could address variability in sensitivity and precision. Moreover, validating the model on larger and more diverse datasets, including those with noisy signals and overlapping AFib episodes, will be critical for its clinical applicability.

## 7. Limitations

Despite the promising performance of our proposed framework, several limitations must be acknowledged:**Data Augmentation and Preprocessing:** Although we employed a range of data augmentation techniques to enhance model robustness, the impact of these augmentations on overall performance was not systematically evaluated. Future studies should explore a wider array of augmentation strategies and quantify their effect, especially under varying noise conditions and artifact levels.**Limited Exploration of Pretraining Approaches:** We did not investigate unsupervised or self-supervised pretraining methods (e.g., masked autoencoders) that could improve feature extraction, particularly in scenarios with limited labeled data. Incorporating such techniques might enhance the model’s generalizability.**Comparative Analysis:** While our model was compared against a baseline 1D CNN, the study did not include a comprehensive comparison with other state-of-the-art architectures for 1D arrhythmia classification trained on our composed dataset. A broader comparative analysis would provide a more complete picture of our model’s relative strengths and weaknesses.**Interpretability and Clinical Integration:** Although we applied Guided GradCAM to shed light on the model’s decision-making process, deep neural networks remain largely “black-box” systems. This inherent lack of transparency can hinder clinical trust and adoption. Further research into more robust interpretability techniques is necessary to fully elucidate the model’s reasoning.**Computational Complexity:** The advanced architecture, while effective, is computationally more demanding than simpler models. This increased complexity could pose challenges for real-time implementation, particularly in resource-constrained settings such as wearable devices or edge computing platforms.

## 8. Conclusions

We have developed a compact 1D ConvNeXtV2-inspired network for the detection of AFib/AFL in single-lead ECG signals. By training on a broad suite of datasets (Icentia11k, CPSC 2018/2021, LTAF, PTB-XL, PCC 2017) and evaluating on MIT-AFDB (0.986 F_1_), MIT-ADB and NST, our model—at just 770 k parameters and 46 MFLOPs per 10 s window—achieved state-of-the-art accuracy while remaining computationally lightweight. Guided Grad-CAM interpretability highlights the network’s focus on P-wave morphology and R–R interval irregularities, underscoring clinical relevance.

Looking ahead, we will

Extend to multi-lead ECG inputs to capture spatial arrhythmic patterns;Incorporate additional arrhythmia classes (e.g., LBBB, RBBB, VT, SVT, AVB);Improve the temporal precision of episode boundaries via refined post-processing or sequence modeling;Explore self-supervised pretraining and domain adaptation to further boost generalization across diverse patient populations and recording conditions.

Our results demonstrate that a carefully designed ConvNeXtV2 architecture can deliver clinical-grade arrhythmia detection in real-time, resource-constrained settings, paving the way for broader deployment in wearable and Holter monitoring applications.

## Figures and Tables

**Figure 1 sensors-25-04109-f001:**
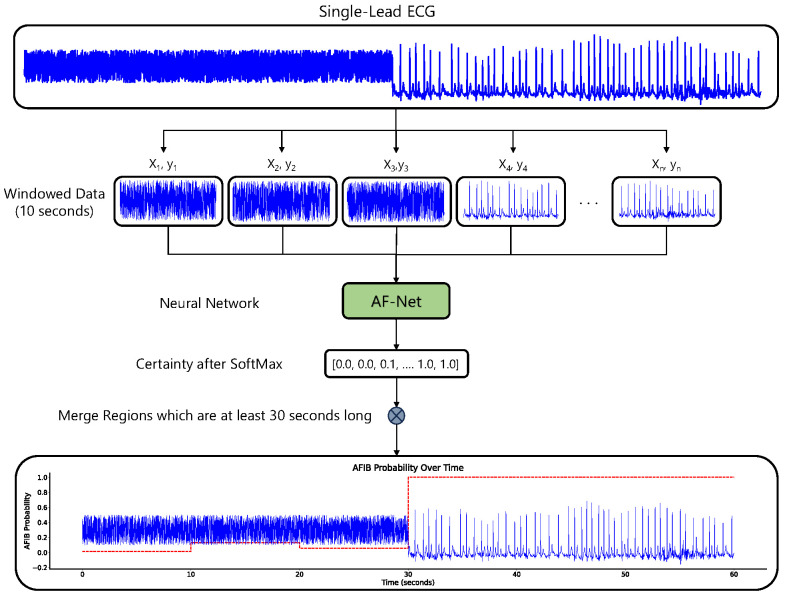
The figure illustrates a method for detecting AFib in single-lead ECG signals using a neural network model.

**Figure 2 sensors-25-04109-f002:**
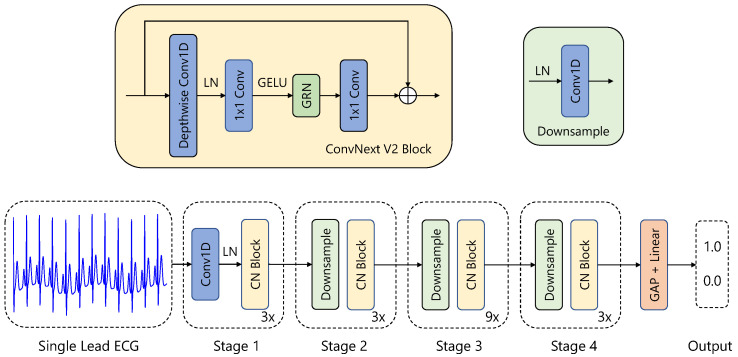
Visualization of our neural network for AFib detection. It is based on a ConvNextv2 [44] architecture adapted for 1D ECG data.

**Figure 3 sensors-25-04109-f003:**
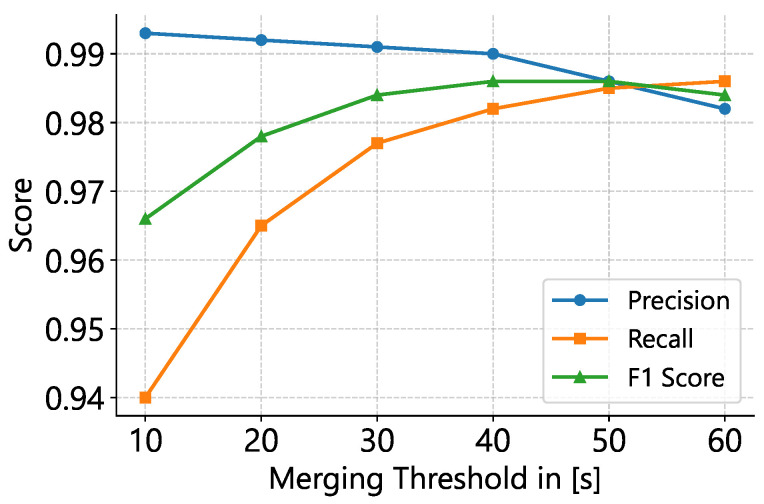
Effect of merging-threshold on classification metrics. Precision, recall, and F_1_–score are plotted for a 10 s window and an AFib probability cutoff of 0.75. The F_1_–optimal threshold lies at 40–50 s, with higher thresholds favoring recall at the expense of precision.

**Figure 4 sensors-25-04109-f004:**
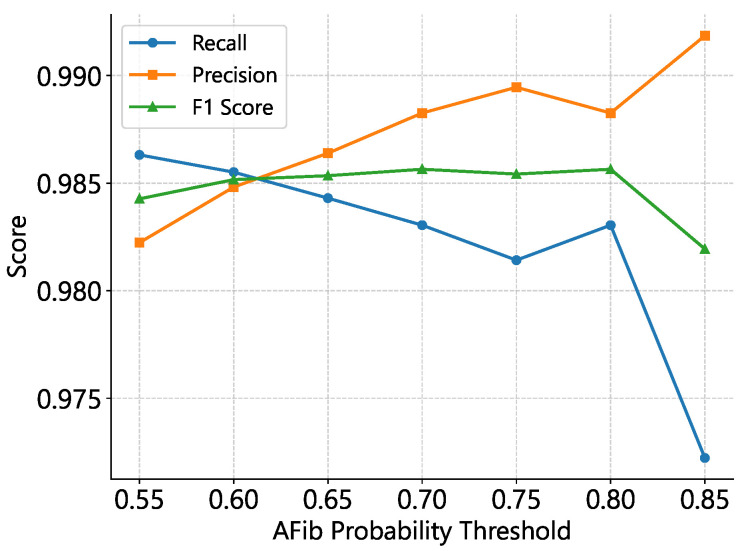
Impact of AFib classification threshold on precision, recall, and F_1_–score for 10 s windows with a 40 s merging gap. The optimal F_1_ range occurs at thresholds of 0.7–0.8, where lower cutoffs favor recall and higher cutoffs favor precision.

**Figure 5 sensors-25-04109-f005:**
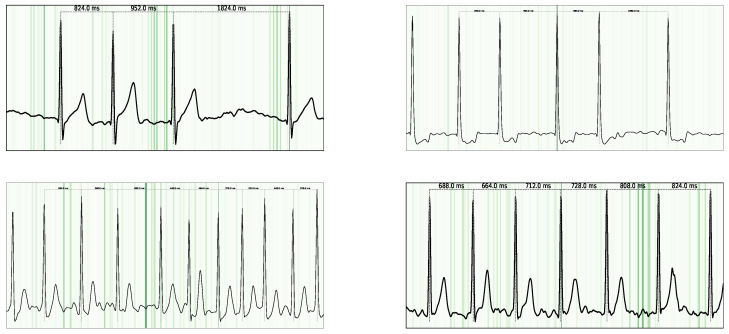
Guided Grad-CAM visualization for AFib examples. Notably, the highest attribution is observed in the P-waves, as their absence is a key criterion for AFib classification.

**Figure 6 sensors-25-04109-f006:**
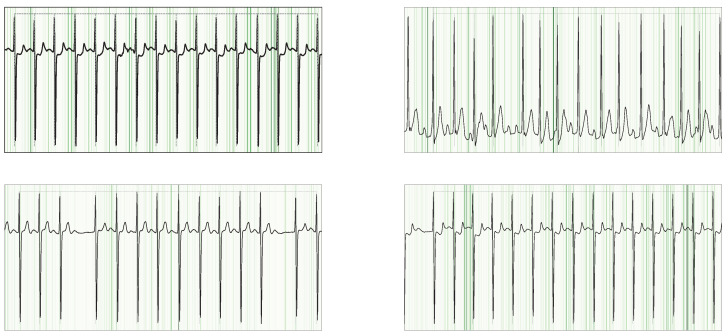
Guided Grad-CAM overlays on NSR signals from the SHDB dataset, highlighting cases with incorrect AFib annotations. Despite the mislabels, the model’s strongest attributions consistently fall on the P-wave regions.

**Table 1 sensors-25-04109-t001:** Comparison of 5 different devices for AFib detection [9].

Manufacturer	Apple	Samsung	Withings	Fitbit	AliveCor
**Version**	Watch 6	Galaxy Watch 3	Scan Watch	Sense	Kardia Mobile
**Sensitivity (95% CI)**	85% (72–94)	85% (72–94)	58% (42–72)	66% (51–79)	79% (64–89)
**Specificity (95% CI)**	75% (67–83)	75% (66–82)	75% (67–83)	79% (70–86)	69% (60–77)
**Inconclusive tracings**	18%	17%	24%	21%	26%
**Preferred Choice**	39%	12%	24%	15%	5%
**Limit of HR interpretation**	50–150	50–120	No information	50–120	50–100
**Battery capacity**	18 h	45 h	720 h	144 h	90 h/2 y
**Price (€)**	449	265	303	244	147

**Table 2 sensors-25-04109-t002:** Summary of studies using PCC 2017 dataset. NSR: Normal Sinus Rhythm; AFib: Atrial Fibrillation; O: Other arrhythmias; N: Noise segments.

Ref.	Classes	Method	$F1_{N}$	$F1_{\mathrm{AFib}}$	$F1_{O}$	$F1_{\mathrm{Mean}}$
Rubin et al. [19]	NSR, AFib, O, N	SQA + DCNN	0.91	0.83	0.72	0.82
Fan et al. [20]	NSR, AFib, O	FRM-CNN	0.93	0.88	0.74	0.85
Zhao et al. [21]	NSR, AFib, O, N	Kalman + DCNN	0.89	0.79	0.72	0.80
Tran et al. [22]	NSR, AFib, O, N	CNN + LSTM	0.90	0.83	0.75	0.80
Cao et al. [23]	NSR, AFib, O, N	2-Layer LSTM	0.91	0.84	0.70	0.82
Nguyen et al. [24]	NSR, AFib, O, N	Stack CNN + SVM	0.93	0.78	0.79	0.83

**Table 3 sensors-25-04109-t003:** Summary of studies using MIT BIH AFDB dataset.

Ref.	Segment Length	Method	Split	Se	Sp/Pr
Henze et al. [25]	21 heartbeats	Generalized linear model	intra-subject	90	95
Liu et al. [26]	30 heartbeats	Normalized fuzzy entropy	intra-subject	98.46	89.85
Faust et al. [27]	100 heartbeats	Bidirectional LSTM	intra-subject	98.46	89.85
Wrobel et al. [28]	21 heartbeats	Linear classifier	intra-subject	95.42	96.12
Pereira and Andreão [29]	10 s	LSTM	inter-subject	91.53	91.00
Martinez et al. [30]	10 heartbeats	CNN	inter-dataset	78.26	91.22
Jahan et al. [31]	20 heartbeats	AdaBoost	inter-subject	87.58	89.27
Teplitzky et al. [6]	60 s ECG	CNN	inter-dataset	97.70	99.70

**Table 4 sensors-25-04109-t004:** Summary of ECG databases used for CNN model training and evaluation.

Database	Number of Records	Type of ECGs	Details
MIT BIH AFDB	23	Two-lead	AFib, AFL, J, N annotations
MIT-BIH ADB	48	Two-lead	AFib and AFL
Icentia11k	541,794	Single-lead	AFib and AFL
CPSC 2018	13,256	Twelve-lead	8 arrhythmia types
CPSC 2021	730 + 706	Twelve-lead	Different AFib types
INCART	75	Twelve-lead	Coronary artery disease examination
PTB and PTB-XL	22,353	Twelve-lead	Cardiac disease examination, 10–120 s per recording
The Georgia 12-lead ECG Challenge	20,672	Twelve-lead	Multiple arrhythmia types, 5–10 s per recording
Chapman–Shaoxing and Ningbo	45,142	Twelve-lead	Multiple arrhythmia types, 10 s per recording
University of Michigan (UMich)	19,642	Twelve-lead	A variety of heart diseases, 10 s per recording
PCC 2017	8528	Single-lead	AFib, NSR, Other, Noise
LTAF	84	Two-lead	Paroxysmal/sustained AFib
CODE-15 SHDB-AF	100 (24 h)	Two-lead	Paroxysmal/sustained AFib, AFL
MIMIC-IV	∼800,000	Twelve-lead	Multiple arrhythmia types, 10 s per recording

**Table 5 sensors-25-04109-t005:** Performance of the ConvNeXt V2 AFib-detection model on the MIT ADB dataset. Overall sensitivity 0.968 and precision 0.944. Records without any true or predicted AFib intervals have been omitted.

Record	Sensitivity	Precision	F_1_ Score	Ref Afib Duration	Pred Afib Duration	Overlap Duration
201	0.993	0.836	0.908	00:10:05	00:12:00	00:10:01
202	0.991	0.953	0.972	00:10:34	00:11:00	00:10:28
203	0.997	0.980	0.989	00:29:29	00:30:00	00:29:23
210	0.997	0.980	0.989	00:29:29	00:30:00	00:29:24
219	0.979	0.878	0.925	00:23:46	00:26:30	00:23:16
221	0.997	0.973	0.984	00:29:16	00:30:00	00:29:11
222	0.593	0.946	0.729	00:08:46	00:05:30	00:05:12
**Overall**	**0.968**	**0.944**	**0.957**	**02:21:29**	**02:25:00**	**02:16:57**

**Table 6 sensors-25-04109-t006:** Performance on the MIT AFIB DB (ConvNeXt V2) using an input window size of 10 s with a merging threshold of 40 s and an AFib class probability threshold of 0.75.

Record	Sensitivity	Precision	F_1_ Score	Ref Afib Duration	Pred Afib Duration	Overlap Duration
4015	0.925	0.298	0.451	00:03:57	00:12:15	00:03:39
4043	0.928	0.911	0.920	02:12:12	02:14:45	02:02:45
4048	0.862	0.972	0.914	00:06:00	00:05:20	00:05:11
4126	0.980	0.986	0.983	00:22:57	00:22:50	00:22:30
4746	0.999	1.000	0.999	05:25:53	05:25:35	05:25:30
4908	0.995	0.997	0.996	00:55:36	00:55:30	00:55:18
4936	0.968	0.992	0.980	08:19:11	08:07:10	08:03:22
5091	0.543	0.943	0.689	00:01:26	00:00:50	00:00:47
5121	0.943	0.972	0.957	06:26:48	06:15:35	06:04:56
5261	0.911	0.910	0.911	00:07:59	00:08:00	00:07:17
6426	0.995	0.991	0.993	09:47:11	09:49:45	09:44:32
6453	0.917	0.987	0.951	00:06:11	00:05:45	00:05:40
6995	0.968	0.986	0.977	04:49:30	04:44:10	04:40:07
7162	1.000	1.000	1.000	10:13:42	10:13:40	10:13:39
7859	0.993	1.000	0.996	10:13:42	10:09:15	10:09:15
7879	1.000	1.000	1.000	06:10:02	06:10:00	06:09:57
7910	0.980	0.996	0.988	01:45:55	01:44:15	01:43:49
8215	0.998	1.000	0.999	08:15:24	08:14:35	08:14:31
8219	0.983	0.916	0.948	02:12:28	02:22:05	02:10:11
8378	0.826	1.000	0.904	02:34:11	02:07:20	02:07:18
8405	1.000	1.000	1.000	07:23:09	07:23:10	07:22:58
8434	0.993	0.956	0.974	00:23:43	00:24:40	00:23:34
8455	0.987	1.000	0.993	07:04:31	06:59:00	06:58:58
**Overall**	**0.982**	**0.990**	**0.986**	**95:01:49**	**94:15:30**	**93:15:52**

**Table 7 sensors-25-04109-t007:** Performance comparison on MIT AFIB and MIT ADB datasets (window merging threshold 40 s, AFib threshold 0.75, window size 10 s).

Model	Sensitivity	Precision	F_1_ Score
**MIT AFIB Dataset**			
ConvNextV2	0.982	0.990	0.986
1D CNN Baseline	0.948	0.992	0.970
**MIT ADB Dataset**			
ConvNextV2	0.982	0.934	0.958
1D CNN Baseline	0.964	0.960	0.962

**Table 8 sensors-25-04109-t008:** Performance metrics for various window sizes and merging thresholds on MIT AFDB.

Window Size (s)	Merging Threshold (s)	Sensitivity	Precision	F_1_ Score
10	10	0.940	0.993	0.966
10	20	0.965	0.992	0.978
10	30	0.977	0.991	0.984
**10**	**40**	**0.982**	**0.990**	**0.986**
**10**	**50**	**0.985**	**0.986**	**0.986**
10	60	0.986	0.982	0.984
20	20	0.968	0.992	0.980
20	50	0.981	0.985	0.983
20	60	0.981	0.984	0.982
30	30	0.970	0.991	0.980
30	50	0.976	0.988	0.982
30	70	0.981	0.982	0.981

## Data Availability

The data presented in this study are available on request from the corresponding author. The data are not publicly available due to privacy reasons.

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
