# Peer review of "Atrial Fibrillation and Atrial Flutter Detection Using Deep Learning"

_sensors, 2025, doi:10.3390/s25134109_

Round 1
Reviewer 1 Report
Comments and Suggestions for Authors
The paper introduces a technically sound and clinically relevant approach to AF detection using deep learning. However, I think substantial revisions are required:
- The abstract and methodology sections could be made more concise and readable. This would help readers from both technical and clinical backgrounds.
- I suggest clearly separating AF and AFL results or revising the title and scope to reflect the true focus of the work.
- The paper reports high sensitivity and precision, but I noticed the test sets contain many samples with no AF, which could inflate precision and specificity. I would recommend including F1 scores and specificity across all datasets for a more balanced evaluation.
- The baseline 1D CNN is a good start, but I believe the paper would benefit from comparison to more competitive or commonly used models in the field. I suggest extending the comparison to include these models to better position the proposed method.
- The decision to merge predicted AFib segments only if they last at least 30 seconds is interesting, but I would like to see a clearer justification. Is there a clinical reason for this threshold? Was its impact on model performance empirically validated?
- The discussion of Guided Grad-CAM is a great addition, but currently feels a bit shallow. I encourage the authors to go deeper into which ECG features were most relevant and how that aligns with clinical knowledge.
- Including model complexity (parameter counts, inference times) would be helpful, especially since the model is intended for real-time or resource-constrained settings.
Author Response
Thank you very much for your valueable comments. We attached the response to your comments and revised our manuscript.

Reviewer 2 Report
Comments and Suggestions for Authors
I have reviewed your paper and identified several areas where improvements are needed. Following, some comments:
- Title: From the title, I expected you to deal with both atrial fibrillation and atrial fluttering, but carefully reading your manuscript, I understand that you concentrate on atrial fibrillation and not on atrial flutter. Atrial flutter is mentioned in the Title, Abstract and Results. For this reason, I suggest you change the title of your manuscript and make it relative to the content.
- Introduction: I recommend rewriting this paragraph; the introduction is a key section for presenting your work. As written, the ideas are not clear, it is not evident what you are exploring or what novelty you are introducing. Since there is already research available on using Holter monitors to detect AF, I suggest updating this paragraph to clearly explain what the reader will read in the rest of the article and how your study differs from existing work.
- Acronym: There has been used more than one acronym for the same term, as atrial fibrillation AFib and AF. Please ensure consistency throughout the manuscript.
- Related Work: You introduced Consumer-graded on the market, but it is not clear how useful it is in achieving your goal. I suggest considering the removal or streamlining of this portion unless you clarify its direct relevance to your methodology or the rationale for your device selection. More generally, if this section is meant to describe the state of the art, it might be helpful to connect these findings more explicitly to your own study to better frame their importance and utility. You need to rewrite Innovative Monitoring Techniques and HRV-based AFIB Detection this because the information you provide is clear but not how to link it with the methodology used.
- Tables and Figures: I suggest you introduce the table or the figure in the text and then it follows the text. This is currently missing, which can confuse the reader and make it harder to follow the content. Also, please rewrite acronyms on the table to make the readability simpler, you can place them in footnotes to improve readability.
- Figure 1: It is not clear how a reader from a single image with no description in the text should understand how it will be used in the methodology. Please take care to make the methodology clear, it should be possible for another researcher that takes it as a reference to reproduce this work.
- Data Preparation: In this section it is explained how you created the data and in line 302 it has written multiple preprocessing algorithms, which are not sufficient to have a clear methodology. May you clarify which algorithms you used? In line 311 is written to meet specific model input requirements. It should be useful to know which are this input requirements are, can you introduce them here or even explain in detail when you introduce the model?
- HDF5 Dataset: It is a combination of the datasets that you introduced, but I think that you have to introduce also PCN202, MIMIC 4 and NST or if they refer to one of the previous databases you introduced, please take care to have consistency.
- Table 5: It needs to be understood how details help in better understand what the table understand. In the text it is needed before the table be explained better why this table is useful and under which criteria the details are achieved.
- Post-Processing: Can you provide a reference for the merging together of the regions? If yes, please provide it in your work.
- Results: This paragraph is called results but what you introduce is the statistical analysis you performed. I recommend making changes in the way to have a paragraph for the statistical analysis and the actual results you obtained.
- Discussion: I suggest a more detained interpretation of the methodology you used and why and how you connect it to your state of the arts.
- Author Contributions: I noticed that a third person has contributed to this work, I suggest his name be included also in the paper.
Author Response
Thank you very much for your valueable review! We have changed our manuscript accordingly and added the full response as pdf.

Round 2
Reviewer 1 Report
Comments and Suggestions for Authors
The authors have addressed all my previous concerns. I have no further comments. Thank you.
Author Response
Thank you for the approval!